# Augmented Reality for Vehicle-Driver Communication: A Systematic Review

**Liam Kettle** * and **Yi-Ching Lee**

Department of Psychology, George Mason University, Fairfax, VA 22030, USA
* Correspondence: lkettle@gmu.edu

**Abstract:** Capabilities for automated driving system (ADS)-equipped vehicles have been expanding over the past decade. Research has explored integrating augmented reality (AR) interfaces in ADS-equipped vehicles to improve drivers' situational awareness, performance, and trust. This paper systematically reviewed AR visualizations for in-vehicle vehicle-driver communication from 2012 to 2022. The review first identified meta-data and methodological trends before aggregating findings from distinct AR interfaces and corresponding subjective and objective measures. Prominent subjective measures included acceptance, trust, and user experience; objective measures comprised various driving behavior or eye-tracking metrics. Research more often evaluated simulated AR interfaces, presented through windshields, and communicated object detection or intended maneuvers, in level 2 ADS. For object detection, key visualizations included bounding shapes, highlighting, or symbols. For intended route, mixed results were found for world-fixed verse screen-fixed arrows. Regardless of the AR design, communicating the ADS' actions or environmental elements was beneficial to drivers, though presenting clear, relevant information was more favorable. Gaps in the literature that yet to be addressed include longitudinal effects, impaired visibility, contextual user needs, system reliability, and, most notably, inclusive design. Regardless, the review supports that integrating AR interfaces in ADS-equipped vehicles can lead to higher trust, acceptance, and safer driving performances.

**Keywords:** autonomous vehicles; automated driving systems; systematic review; augmented reality; heads-up display; human–computer interaction; trust; situational awareness; user experience

## 1. Introduction

The development of automated driving system (ADS) capabilities and technologies equipped in vehicles has been expanding over the past decade, with driving control shifting towards the ADS. Recently, ref. [1] updated its taxonomy for the six automation levels ranging from level 0 (no driving automation) to level 5 (full driving automation). At levels 2 and 3, the automation features assist in lateral and longitudinal control (i.e., lane keeping and adaptive cruise control, respectively). When something goes wrong such as if the road condition exceeds the ADS capabilities, then vehicle operation will fallback to the human driver. In many cases, the ADS will issue a take-over request (TOR) whereby the ADS alerts the driver to fully resume manual control in a short time span. When level 3 and above ADS features are engaged, there is reduced need for constant driver monitoring of the road environment until the ADS is capable of full driving automation (level 5). Given the reduced need for driver oversight, the driver may engage in non-driving related tasks which can lead to decreased driving performance [2–4] and increased crash risk [5]. When a driver shifts visual attention away from the road environment and toward a non-driving related task, they lose situation awareness of critical road cues needed to update their dynamic mental model of the driving context [6]. Reduced situation awareness during TORs places the driver in potentially dangerous driving situations whereby delayed or inappropriate reactions while discerning the driving scene can lead to dangerous outcomes [7–9]. However, at higher levels where TORs are less prevalent

(i.e., levels 4 and 5), driving performance is a less crucial factor, rather, drivers' trust and acceptance of the ADS-equipped vehicle are more important factors for the adoption of vehicles equipped with high or full driving automation features [10].

One facet of ADSs that can mitigate reduced situation awareness as well as improve perceptions of ADS-equipped vehicles is through transparent vehicle-human communication. Past research suggests that appropriate expectations of the systems capabilities as well as understanding how the system performs and predicts future behavior can improve trust [11,12]. Ref. [13] found that drivers desired vehicle interfaces that communicate information relevant to the ADS's situation awareness of the road environment (what the system perceives) and behavioral awareness (what actions the system will take). Similar desires are found for ADSs that clearly convey information relevant to oncoming critical situations, the ADS's decision making and its actions [14–18].

Previous research has evaluated various strategies that communicate the ADS's detection of potential hazards or its intended actions. More specifically, communication strategies have included visual [19,20], audible [21–24], olfactory [25], haptic [26–28], and multimodal [7,29] interfaces. Additionally, researchers have evaluated the communication strategies of embodied agents such as a NAO robot with speech features [30,31] or directional eye-gaze of three social robots to alert drivers of potential critical cues in the driving environment [32,33]. However, many of these communication avenues are ambiguous or allocate visual attention outside of the road environment which can lead to potentially fatal outcomes. Instead, augmented reality (AR) can be utilized to communicate road elements and ADS actions without allocating visual attention away from the driving environment.

AR represents a component of mixed-reality, in which the virtual and real world are merged [34]. More specifically, virtual images are superimposed on the real world, enriching an individuals' sensory perception [35] of reality. Currently, AR applications are strongly utilized in many areas within the automotive industry including vehicle assembly, design, maintenance and manufacturing [36]. Additionally, in-car AR systems are utilized to communicate road cues to the driver through head-up displays (HUDs). HUDs convey visual information (e.g., road cues including pedestrians, vehicles, and signs) in the drivers' field of view. Currently, two main modalities are used to present AR visualizations. First, AR visualizations can be presented through optical see-through HUDs (e.g., [37,38]) which are transparent devices that occupy a small area of the driving field of view; secondly, through windshield displays in which AR visualizations can occur anywhere on the drivers' forward field of view (e.g., [39,40]). Typically, information is communicated to the driver by highlighting certain road cues already present in the environment or by displaying additional information onto the environment [41].

Through AR visualizations, the ADS can communicate its intention in detecting road elements and convey future ADS actions. Accordingly, communicating transparent driving-related information can improve individuals' situation awareness (i.e., allocation of visual attention and driving performance) of the driving environment [37,38,42]. Furthermore, communicating to drivers what the ADS "perceives" can improve overall trust and acceptance [18], Ref. [43] while dynamically calibrating appropriate expectations of the ADS, which in turn can foster better adoption of ADSs. Currently, there are various in-vehicle AR designs that communicate a broad range of information; however, these diverse designs are generally evaluated independent of other visualizations making it difficult for researchers to integrate or compare results. Therefore, current AR designs should be systematically reviewed to identify which visualizations are more prominent in AV applications for information communication and understand potential gaps in the literature for future directions.

### 1.1. Relevant Past Reviews

There have been a number of past reviews: Ref. [44] systematically reviewed AR usability studies across a broad range of applications, one of which includes automotive applications. The authors summarized the high-level contributions of influential studies

and provided insight to how these studies further developed the AR landscape. Ref. [36] then reviewed AR articles in the automotive industry across four classifications: review papers (i.e., those summarizing existing literature), technical papers (i.e., development of algorithms or software/hardware for AR systems), conceptual papers (i.e., propositions of new concepts to adopt), and application papers (i.e., development and testing in simulated or real environments). Within AR applications, ref. [36] identified five contexts of AR use areas including assembly, design, manufacturing, maintenance, and 'in-car' systems, and presented general potential solutions, benefits, and the technological challenges. Ref. [45] presented an overview on the existing methodological approaches of user studies on automated driving. Specifically, they reviewed the various constructs evaluated across user studies (e.g., safety, trust, acceptance, workload) and how these constructs were measured, but did not discuss the contexts of when these measures were applied. Refs. [46] and [47] both presented key areas of AR usage in ADS-equipped vehicles such as navigation, safety, driver trust, and gamification. Ref. [47] provided an in-depth high-level review while ref. [46] additionally conducted a conference workshop. Both discussed the opportunities and challenges for AR applications for drivers, pedestrians, and other road users but did not evaluate impact of AR designs for these areas. Ref. [48] reviewed the design of external human–machine interfaces for vehicle-pedestrian communication comparing the performance measurement distinctions between monitor- and VR-based experimental procedures. However, these evaluations did not include 'in-vehicle' interface evaluations. Finally, ref. [49] reviewed the opportunities and challenges of incorporating in-vehicle AR applications across interface type (e.g., head-up display, head-mounted display, and device) and location (screen-fixed vs. world-fixed display). However, ref. [49] only focused on the perceptual and distraction issues relating to these AR applications. While these prior reviews provided valuable information about key application areas of AR in the automotive industry, the emerging challenges and opportunities of in-vehicle and vehicle-to-pedestrian AR communication displays, they did not include the variety of AR visualizations for vehicle-driver communication. Therefore, the current review focused on the specific designs of in-vehicle AR visualizations that communicate either road elements (e.g., pedestrians, vehicles, signage) or the ADS's actions and decision-making to the driver across levels of automated driving.

### 1.2. Aim of the Study

The overall objective of the review was to identify the trends in AR visualizations for in-vehicle vehicle-driver communication in ADS-equipped vehicles from published research articles between 2012 and 2022. More specifically, the review aggregated AR designs evaluated by participant research and identified the leading visualizations for communicating object detection and ego vehicle actions. The main contributions of the review include (a) providing an overview of the meta-data and methodologies commonly utilized in this research space, (b) aggregating AR designs for information communication and their impacts across driving performance, eye-tracking, and subjective measures, (c) identifying the gaps and challenges in the current research, and (d) providing recommendations for potential future directions for AR research within this domain.

## 2. Materials and Methods

### Paper Selection

To find relevant research articles, five well-known online research databases were used: ACM digital library, Google Scholar, IEEE Xplore, ScienceDirect, and Web of Science. Database searches occurred during the summer of 2022 using a modified version of the PRISMA [50] guideline. PRISMA helps authors provide consistent standards for reporting reviews and meta-analyses. The following search method modified the PRISMA guidelines by (a) non-reporting of duplicated articles as any duplicates found was skipped and automatically excluded, and (b) article searching continued in a database until 100 consecutive articles were deemed irrelevant to the topic (as outlined in [51]). The objective of the

search was to identify research articles that statistically evaluated AR visualizations that communicate the ADS's actions, decision-making, or environmental features (e.g., vehicle or pedestrian detection) to the human driver. For each of the databases, a BOOLEAN search term was used to capture a broad range of articles, sorted by relevance–(e.g., "head up display" OR "augmented reality") AND ("autonomous vehicle" OR "self driving vehicle"). The terms "autonomous vehicle" and "self driving vehicle" were used initially as [1] recently updated its terminology, thus, previous studies would not have incorporated the updated terms. A secondary search was then conducted including the term "automated driving system" for articles published since 2021. The title and abstracts were screened first for relevancy; if the title and abstract presented confusion, then the article was read in full. Figure 1 summarizes the PRISMA flow chart and the search results. The inclusion criteria included:

- Peer-reviewed, original full research
- Clearly describes the AR design and its target features
- Graphical representation of the displays
- User testing and evaluative comparison of AR display designs using statistical analysis
- Originally published in the past decade (2012–2022)
- In English
- Exclusion: Work-in-progress articles
- Exclusion: AR that communicates to actors outside the vehicle

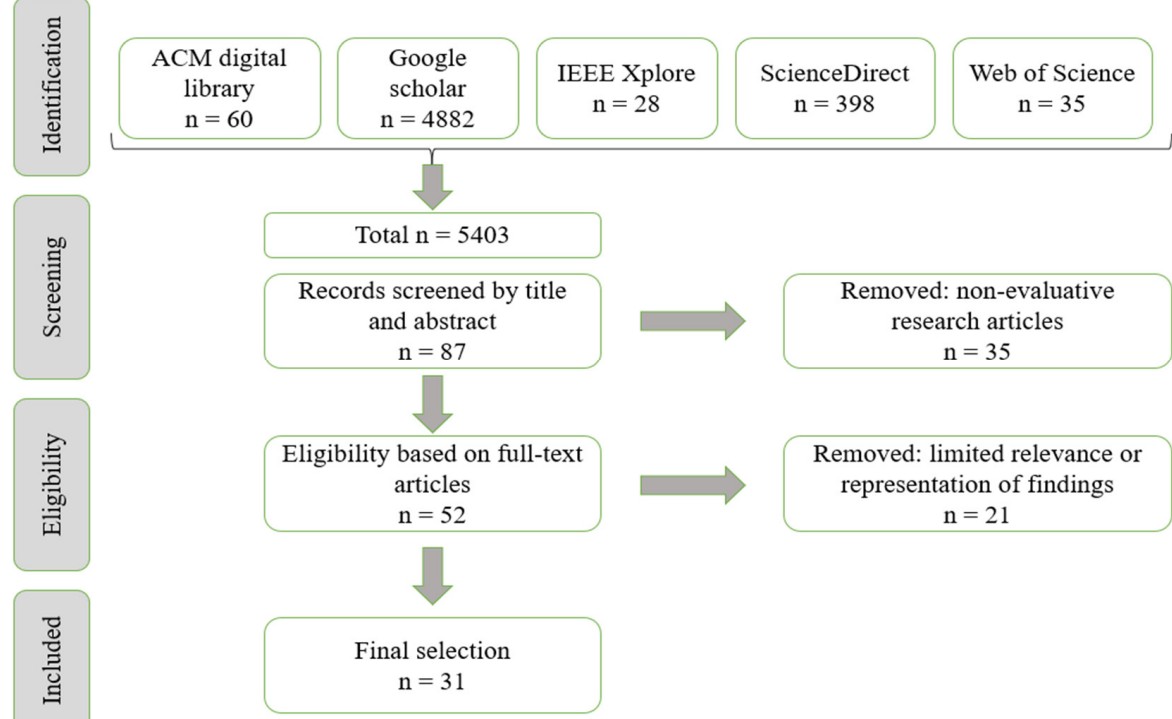

**Figure 1.** PRISMA flow chart depicting the article search process.

The search process identified a total of 5403 potential results extracted from the five library databases. Articles were first screened for relevance to in-vehicle AR designs by title and abstract which identified 87 relevant articles. These articles were screened again for evaluative methods by title and abstract and 35 were removed due to non-evaluative articles (e.g., book chapters, theoretical reviews). The remaining 52 were read in full and screened for eligibility according to the inclusion and exclusion criteria. A further 21 articles were removed due to limited relevance or representation of findings. Thus, 31 articles were included in the final selection.

Articles published in conference proceedings were considered peer-reviewed if the conference organization explicitly states that published research adhere to a peer-review process. If no statement could be found, then the article was excluded. Although the search terms were focused on ADS-equipped vehicles, identified research articles that did not utilize vehicles equipped with any ADS features were still deemed relevant as they provided visualizations that may be useful for vehicles operating with ADS features.

Although the search strived to extract as many relevant articles as possible using a wide range of venues, we acknowledge that additional articles may have been missed due to omitted venues. Furthermore, the search terms used were carefully chosen to capture a wide berth of research articles; however, there may be articles that do not includes the chosen keywords when describing AR visualizations in ADS-equipped vehicles.

## 3. Results

In total, 31 relevant articles were reported. Table 1 displays a summary of the relevant articles. First, the section will start with descriptives of high-level article features such as articles over time, origin, types and design, then will shift into reviewing the various AR visualization displays presented across studies.

### 3.1. High-Level Overview of Reviewed Articles

This section presents an overview of high-level descriptive information across the articles. First, this section describes the publication of articles over time, origin, type, and design. Next, information relevant to participants, recruitment, and accessibility, followed by the subjective and objective types of data collected. Finally, this section finishes with a summary of the levels of automation across the articles and the descriptive information of the various AR designs and communicated information identified.

#### 3.1.1. Articles over Time

Over the past decade, 22 (71%) research articles were published in the last five years (Figure 2). However, this number is inclined to increase as the results do not account for any possible future articles published in 2022. Overall, 15 (48%) were journal publications and 16 (52%) were conference articles. Taking 2018 as an exception, the number of journal publications is relatively consistent of one to two articles per year. In contrast, the number of conference proceeding articles increased in the past three years with 2021 seeing eight conference articles published.

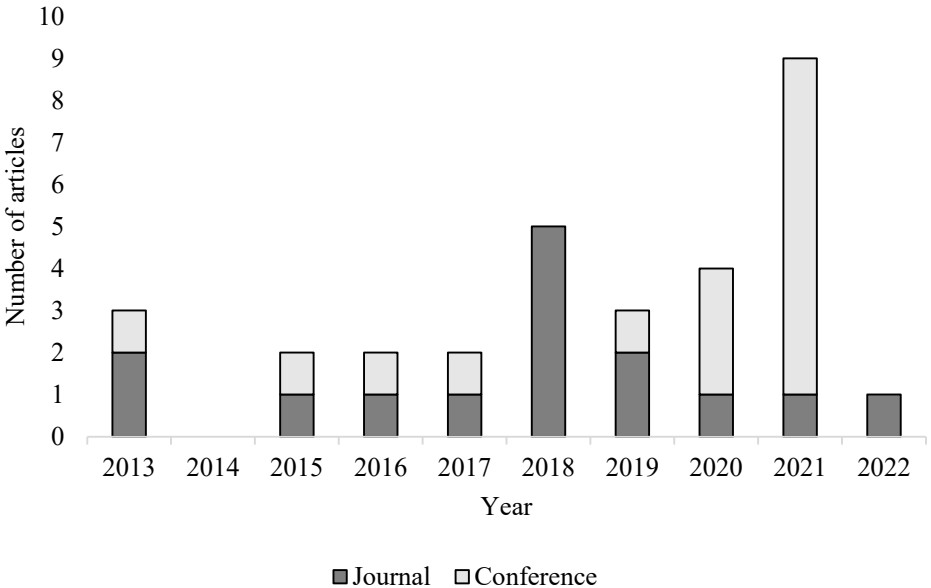

**Figure 2.** Article publication type over time.

**Table 1.** Summary of article descriptives, study design, and display design.

| Reference | Year | Publication Type | Topic | N Participants | Mean Age | Age Range | Gender M:F (Other) | ADS Level | Study Design | Display Mode | Display Design | Displayed Information |
|---|---|---|---|---|---|---|---|---|---|---|---|---|
| Colley et al. [39] | 2021 | Conference | Communicating hazard detection | 20 | 32.05 | N/A | 10:10 | HAV | Within | WSD | Warning symbols; lightband; bounding circle; bounding circle & symbol | Hazard detection; pedestrian detection; vehicle detection |
| Colley et al. [52] | 2020 | Conference | Communicating pedestrian recognition and intent | 15 | 25.33 | N/A | 11:4 | HAV | Within | WSD | Symbol & icons | Pedestrian detection & intention |
| Colley et al. [53] | 2021 | Conference | Communicating hazard detection | Pilot: 32 Main: 41 | Pilot: 27.06 Main: 27.63 | N/A | Pilot: 23:9 Main: 20:21 | HAV | Within | WSD | Solid highlight | Dynamic (vehicles; pedestrians); static (road signs) |
| Colley et al. [54] | 2021 | Conference | Takeover performance | 255 | 31.46 | N/A | 139:86 | 2 | Between | WSD | Warning symbols; bounding boxes | Hazard detection |
| Currano et al. [55] | 2021 | Conference | Situation Awareness via AR | 298 | 35.1 | N/A | 156:139 | N/A | Mixed | Simulated optical HUD | Circle; highlighting; arrow | Hazard detection; intended route |
| Detjen et al. [56] | 2021 | Conference | Intended route navigation | 27 | 21.74 | N/A | 24:3 | Study 1: 3–5 Study 2: 2 | Within | WSD | Symbols and icons; arrows | Intended route; vehicle detection |
| Du et al. [18] | 2021 | Conference | Acceptance of alert system | 60 | 24 | 18–45 | 35:25 | 3 | Mixed | WSD | Bounding boxes; warning symbols; arrows | Hazard detection; road signs |
| Ebnali et al. [57] | 2020 | Conference | Communicating system status and reliability | 15 | 26.02 | 21–34 | 7:8 | 2 | Mixed | WSD | Lane marking | Automation reliability |
| Eyraud et al. [41] | 2015 | Journal | Visual Attention Allocation | 48 | 34 | 21–53 | 27:21 | 0 | Mixed | WSD | Solid highlight | Intended route; road signs |
| Faria et al. [58] | 2021 | Conference | Takeover performance | 8 | 23.56 | 18–30 | N/A | 2 | Within | WSD | Bounding boxes | Pedestrian detection; Vehicle detection |
| Fu et al. [59] | 2013 | Conference | Improving safety through AR | 24 | Young: 20 Older: 65 | 18–75 | 14:10 | 0 | Mixed | Projected on screen | Colored blocks; lines | Merging traffic; braking vehicles |
| Gabbard et al. [37] | 2019 | Journal | Intended route navigation | 22 | Males: 20.3 Females: 20.4 | N/A | 13:9 | 0 | Within | Optical HUD | Arrows | Intended route |

**Table 1.** *Cont.*

| Reference | Year | Publication Type | Topic | N Participants | Mean Age | Age Range | Gender M:F (Other) | ADS Level | Study Design | Display Mode | Display Design | Displayed Information |
|---|---|---|---|---|---|---|---|---|---|---|---|---|
| Hwang et al. [60] | 2016 | Journal | Risk perception using AR | 28 | AR: 32.27 Control: 39.08 | N/A | 28:0 | 0 | Between | Optical HUD | Lines | Pedestrian detection; Vehicle detection |
| Jing et al. [42] | 2022 | Journal | Takeover performance | 36 | 22.7 | 18–31 | 24:12 | 3 | Between | WSD | Arrows; highlighting | Pedestrian detection; Vehicle detection |
| Kim & Gabbard [61] | 2019 | Journal | Visual Distraction by AR interfaces | 23 | 21.4 | N/A | N/A | N/A | Within | WSD | Bounding boxes; shadow highlighting | Pedestrian detection |
| Kim et al. [38] | 2018 | Journal | Communicating hazard detection | 16 | 42 | N/A | N/A | 0 | Within | Optical HUD | Shadow highlighting | Pedestrian detection |
| Lindemann et al. [62] | 2018 | Journal | Situation Awareness via AR | 32 | 27 | 19–58 | 24:8 | 4–5 | Within | WSD | Symbols | Automation reliability; hazard detection; intended route |
| Lindemann et al. [40] | 2019 | Conference | Takeover performance | 18 | 25 | N/A | 10:8 | 3 | Within | WSD | Arrows; highlighting | Hazard detection; intended route |
| Merenda et al. [63] | 2018 | Journal | Location identification of parking spaces and pedestrians | 24 | 27.35 | 18–40 | 17:7 | 0 | Within | Optical HUD | Warning symbols | Object location |
| Oliveira et al. [64] | 2020 | Journal | Communicating navigation and hazard detection | 25 | N/A | N/A | 21:4 | 4 | Within | WSD | Warning symbols; lane marking | Hazard detection; intended route |
| Pfannmuller et al. [65] | 2015 | Conference | Intended route navigation | 30 | 32.9 | 22–52 | 23:7 | N/A | Within | Optical HUD | Arrows | Intended route |
| Phan et al. [66] | 2016 | Conference | Communicating hazard detection | 25 | N/A | 21–35 | 21:4 | 0 | Within | Simulated optical HUD | Bounding boxes; warning symbols | Pedestrian detection |
| Rusch et al. [67] | 2013 | Journal | Directing attention | 27 | 45 | N/A | 13:14 | 0 | Within | WSD | Converging bounding box | Pedestrian detection; vehicle detection; road signs |

**Table 1.** *Cont.*

| Reference | Year | Publication Type | Topic | N Participants | Mean Age | Age Range | Gender M:F (Other) | ADS Level | Study Design | Display Mode | Display Design | Displayed Information |
|---|---|---|---|---|---|---|---|---|---|---|---|---|
| Schall et al. [68] | 2013 | Journal | Hazard detection in elderly | 20 | 73 | 65–85 | 13:7 | 0 | Within | WSD | Converging bounding box | Pedestrian detection; vehicle detection; road signs |
| Schewe & Vollrath. [69] | 2021 | Journal | Visualizing vehicle's maneuvers | 43 | Males: 37 Females: 34 | N/A | 35:8 | 2 | Within | WSD | Symbol; lane marking | Speed changes |
| Schneider et al. [70] | 2021 | Conference | UX of AR transparency | 40 | 24.65 | N/A | 26:14 | 5 | Mixed | WSD | Bounding boxes; lane marking | Hazard detection; intended route |
| Schwarz & Fastenmeier. [71] | 2017 | Journal | AR warnings | 81 | 31 | 20–54 * | 70 *:18 * | 0 | Between | Optical HUD | Warning symbols; arrows | Hazard detection |
| Schwarz & Fastenmeier. [72] | 2018 | Journal | AR warnings | 80 | 31 * | 22–55 | 70:10 | 0 | Between | Optical HUD | Warning symbols | Hazard detection |
| Wintersberger et al. [73] | 2017 | Conference | Trust in ADSs through AR | 26 | 23.77 | 19–35 | 15:11 | 5 | Within | WSD | Symbols & icons | Vehicle detection |
| Wintersberger et al. [43] | 2018 | Journal | Trust in ADSs through AR | Study 1:26 Study 2:18 | Study 1:23.77 Study 2:24.8 | Study 1: 19–35 Study 2: 19–41 | Study 1:15:11 Study 2:12:6 | 5 | Within | WSD | Symbols & icons; arrows | Vehicle detection; intended route |
| Wu et al. [74] | 2020 | Conference | Communicating hazard detection | 16 | N/A | N/A | 12:4 | 2 | Within | WSD | Bounding boxes | Pedestrian detection; vehicle detection |

Note: * the article described participant demographics before the exclusion of participants; ADS = Automated driving system; HAV = Highly autonomous vehicle; WSD = Windshield display; HUD = Heads-up display.



### 3.1.2. Article Origin

To provide a global picture of the origin of articles, the affiliation of the first author was considered the reference point for article location. Overall, 17 (55%) of the publications emerged from Europe, 12 (39%) from North America, and 2 (6%) from Asia. Breaking it down further, 14 (45%) articles emerged from Germany, 12 (39%) from the United States, 2 (6%) from France, and 1 (3%) each from China, South Korea, and United Kingdom.

### 3.1.3. Article Type and Design

All but five (16%) of the articles included one study. Two articles conducted an initial study–one utilized a survey, the other a design workshop–exploring individuals' preferences and priorities regarding possible AR interface elements, followed by the experimental studies. One article conducted a pilot study before their main evaluative study. Finally, two articles reported two studies each: one article using separate samples, one article using the same sample. Experimental design strongly favored within-subjects methodology ($n = 20$, 65%), with five (16%) using a between-subjects methodology, and six (19%) using a mixed approach. The majority of studies were conducted in laboratory settings ($n = 24$, 77%), with four (13%) conducted online, two (6%) conducted in a controlled outdoor setting, and one (3%) conducted in an indoor controlled setting. Of the research conducted in laboratory settings, seven utilized real-world footage; for the online research, two utilized simulator-created footage, one utilized real-world footage, and one article utilized real-world footage for their pilot study then simulator-created footage for their main study. The study locations indicates that most conclusions stem from safe and controlled driving situations possibly reducing generalizability to naturalistic settings. Although three articles conducted research in indoor and outdoor settings, these were once again in controlled environments where the risk-free nature and lack of extraneous road actors (i.e., pedestrians, moving vehicles) possibly reduces the generalizability again. However, given that the technology to integrate AR communication displays are still developing, these articles are a good start at identifying display design features that drivers prefer.

### 3.1.4. Participants

Analysis of the participant information revealed that the number of participants ranged from 8–298 with a median of 26 participants. Only three (10%) of articles failed to report gender distribution. However, one article reported gender distribution before the exclusion of participants. Only one (3%) article recruited participants who did not self-identify as male or female. Males were generally always recruited more with 24 (77%) articles having more male participants compared to three (10%) recruiting more females, and one (3%) recruiting an equal distribution. However, articles with more female participants were only greater in number by one participant compared to males. Nineteen (61%) articles described the source of participant recruitment. Of these 19 articles, eight recruited university students, five recruited university staff, five recruited employees working for vehicle manufacturing companies, four recruited from social media (e.g., Facebook), three recruited from mailing lists or flyers, two recruited from the general population, one recruited from a research facility, one recruited from an internal database, and one recruited from word-of-mouth. Articles could have more than one recruitment method. Only one article detailed the ethnicity of their participants. Only two articles recruited vulnerable individuals (i.e., elderly individuals). Examining age, 28 (90%) articles reported the mean age, 26 (84%) reported the standard deviation of age, and 15 (48%) reported the age range of participants. Across the board, the median average age was 27 years old. It should be noted that two articles reported the mean age for each gender without reporting the mean age for the overall sample. Additionally, one study reported the age statistics prior to removing excluded participants. These analyses clearly demonstrate that articles typically comprise of healthy young males and lacks recruitment diversity, especially for non-binary participants and vulnerable or neurodivergent populations for a more representative sample of the population.

### 3.1.5. Accessibility

Six (19%) articles addressed accessibility to some degree. Of these, only two articles recruited elderly individuals who are considered a vulnerable population. Additionally, one article mentioned excluding individuals who were colorblind. Five of the six articles addressed accessibility in their discussion, though these typically stated how the observed results from young healthy populations may differ to other population groups (i.e., elderly individuals). Research should focus on collecting data from various individuals including those from vulnerable populations or neurodiverse individuals. By recruiting these individuals, research can step towards inclusive design and improve transport accessibility.

### 3.1.6. Metrics

Both subjective and objective measures were utilized across the board. Of the articles, nine (29%) reported subjective data only, eight (26%) reported objective data only, and 14 (45%) reported both subjective and objective measures. Table 2 reports the main subjective measures administered and the relevant formal questionnaires. Additionally, eight (26%) administer open-ended feedback questions, three (10%) conducted interviews post-experiment, and one (3%) included minor questionnaires but did not state where they originated. Custom questionnaires included self-created questions that the researchers administered which included, but not limited to, items such as preference, comfort of use, comprehension, attractiveness, trust, navigation, safety, well-being, and intuitiveness. As these differed across articles, custom questionnaires were not analyzed further. Indicated from the tables, there is a large range of scales used to assess participants' subjective perceptions. Although acceptance was measured more often than the other scales, it had a greater number of different scales administered. Trust, user experience, and situation awareness measures each had two administered scales. However, one article that assessed trust administered both scales to examine different facets that comprise overall trust. Thus, the Trust in Automation scale is clearly the dominant trust scale for AR evaluative research. Furthermore, the NASA-TLX seems to be the uncontested measure of subjective workload.

Across the board, there seems to be a favorable trend supporting AR interfaces through the subjective measures. Trust tends to be consistently higher for vehicles integrated with AR interfaces compared to those without, with one article indicating no difference in trust [53]. However, the lack of difference could be due to the participants recognizing limitations and flaws in the recognition of objects. Participants generally perceived the AR interface as having higher usability scores compared to vehicles without AR displays, though conflicting findings were found when comparing variations of specific designs. AR presence typically did not increase perceived cognitive load. Mixed findings were found for acceptance and perceived situation awareness. Generally, there were either similar or increased levels of acceptance between presence and absence of AR interfaces. Finally, AR interfaces were found to increase, decrease, or show no differences compared to no AR interface for situation awareness. Although a positive trend can be suggested, there were subjective differences in design variations and their information contexts which are summarized in Section 3.2.

A summary of objective measures is shown in Table 3. As indicated, roughly half of the studies measure some form of driver behavior, with 10 of the 16 articles assessing braking performance such as braking response time. Approximately one third of the articles also integrate eye-tracking measures with nearly all assessing some form of gaze time such as percentage dwell time. Here, there is a distinction made between gaze time and gaze response. Gaze time more focused on how long an individual fixates on any areas of interests, whereas gaze response more related to the time a driver takes to shift their gaze to a target area after a trigger event. Although driving behavior and eye-tracking are prominent across objective measures, five (16%) specifically measure individuals' situation awareness through the response accuracy of verbal questions. Additionally, five studies utilized non-driving reaction time measures such as button pressing or verbal response as other means to evaluate AR visualizations.

**Table 2.** Summary of administered subjective measures.

| Construct | Name of Subjective Measure | N (%) |
|---|---|---|
| Acceptance | | 11 (35) |
| | Van der Laan Acceptance scale | 4 (13) |
| | Technology Acceptance Model | 4 (13) |
| | AttrakDiff Questionnaire | 2 (6) |
| | Autonomous Vehicle Acceptance Model | 1 (3) |
| Trust * | | 7 (23) |
| | Trust in Automation | 7 (23) |
| | Trust on Adopting an Autonomous Vehicle | 1 (3) |
| User Experience | | 7 (23) |
| | System Usability Scale | 4 (13) |
| | User Experience Questionnaire | 3 (10) |
| Situation Awareness | | 5 (16) |
| | Situation Awareness Global Assessment Technique | 2 (6) |
| | Situation Awareness Rating Technique | 3 (10) |
| Workload | | 5 (16) |
| | NASA-TLX | 5 (16) |
| Affective Driving | | 2 (6) |
| | Self-Assessment Manikin | 2 (6) |
| Driving Behavior | | 2 (6) |
| | Driving Behavior Determinants Questionnaire | 1 (3) |
| | Multidimensional Driving Style Inventory | 1 (3) |
| Anxiety | | 1 (3) |
| | STATE Anxiety Questionnaire | 1 (3) |
| Custom Questionnaire | – | 17 (55) |

Note: N = total number of articles that included the assessment. % = percentage of articles that included the assessment. * one study administered two trust scales so scale number and total number differ by 1.

Overall, presenting AR displays tends to improve driving performance and significantly impacts drivers' visual attention in different ways. Compared to no visual information, AR interfaces tend to improve takeover response time and performance quality with less aggressive braking behaviors and reduces the number of collisions. Additionally, one article indicates that an AR interface induces different patterns of driving behaviors for younger and older drivers. For eye-tracking metrics, findings tend to indicate better critical object identification with lower number of glances or areas of interest scanned before takeover, and reduced effect of visual distraction of road objects. Across the articles though, researchers tend to suggest that AR communication may be less effective for highly salient objects (i.e., vehicles). Although AR interfaces seem to improve the efficiency of visual allocation to critical elements, ref. [57] found drivers to increase their gaze at a phone display. Further, there was suggestion that AR interfaces may direct attention away from uncued road elements. However, no differences were observed via the custom question-response situation awareness measures nor for heat-rate variability though only one article examined this factor. Finally, there was a general trend that AR interfaces improved button pressing response rate and time. Regarding the button press or the custom situation awareness measures, caution is required when interpreting these results due to the possible lack of generalizability or validity of these measures. Similar to the subjective measures, there

were identified differences in design variations for objective outcome which is summarized in Section 3.2.

**Table 3.** Summary of administered objective measures.

| Construct | Objective Measure | N (%) |
|---|---|---|
| **Driving Behavior** | | **16 (52)** |
| | Braking performance | 10 (32) |
| | Takeover performance | 5 (16) |
| | Headway performance | 3 (10) |
| | Collisions | 3 (10) |
| | Lateral performance | 3 (10) |
| | Longitudinal performance | 2 (6) |
| **Eye-tracking** | | **11 (35)** |
| | Gaze time | 8 (26) |
| | Gaze frequency | 5 (16) |
| | Gaze response | 4 (13) |
| | Gaze angle | 1 (3) |
| **Reaction Time** | | **5 (16)** |
| | Button pressing | 4 (13) |
| | Verbal response rate | 1 (3) |
| **Situation Awareness** | | **5 (16)** |
| | Question-response accuracy | 5 (16) |
| **Physiological** | | **1 (3)** |
| | Heart-rate variability | 1 (3) |

Note: N = total number of articles that included the assessment. % = percentage of articles that included the assessment. Bold indicates overall measure and total count. Articles can include multiple items from each measure.

### 3.1.7. Experimental Procedure and Analysis

Various experimental procedures were conducted across the articles and were classified into four main types of comparisons–alert modality, display modality, control x design, and design x design. The latter two were further divided into comparisons with a single design (i.e., control/AR design) or with two or more visual designs (e.g., control/2+ AR design). A summary table of the experimental procedures and the additional independent variables considered are shown in Table 4. Most notably, 20 studies focused on comparing various AR interfaces to a control group (i.e., presented with no information); 12 of these compared only one design to a control group while eight compared a control group to two or more designs. Across these 20 studies, 10 factored additional independent variables into their procedures. For the seven studies that focused on comparing designs, three evaluated two designs with critical situation and visibility as further independent variables; four evaluated three or more designs with distance and urgency as additional variable considerations. For display modality, four studies compared combinations of control, tablet, HDD (e.g., animations or forward camera feed), or AR interface with two studies evaluating different visual designs and one study comparing display modality in different driving scenarios. Finally, two studies compared visual (AR interface) and auditory alerts with information type (e.g., specific, unspecific) as additional factors.

An extensive range of statistical analyses were performed across the articles. A summary of which analyses were performed for the corresponding experimental procedure is shown in Table 4. Across all articles, the parametric ANOVA was performed the most ($n = 18$, 58%). Breaking the ANOVA into its separate forms, there were four (13%) articles that performed one-way ANOVAs, two (6%) performing two-way ANOVAs, six (19%) performing repeated measure ANOVAs, two (6%) performing mixed ANOVAs, and four (13%) performing two-way repeated measure ANOVAs. Following ANOVAs, seven (23%) articles performed *t*-tests for their main analysis. Two (6%) articles performed chi-square tests–one using Pearson's method and the other using Durbin's. The Friedman's test and

Wilcoxon Signed Rank test were each performed by six (19%) articles. The Kruskal–Wallis test, Mann–Whitney test, the non-parametric Analysis of Variance (NPAV), and Pearson correlations were each performed by two (6%) articles. Five (16%) articles performed linear mixed effect models with cumulative link model, logistic regression, and multiple regression analyses being performed by one (3%) article each. Finally, eighteen (58%) articles conducted post hoc analysis. All but one post hoc analysis involved pairwise *t*-test comparisons–the one performed McNemar's test instead. Across these post hoc comparisons, 11 (35%) used Bonferroni corrections, four (13%) used Tukey adjustments, and Fisher's LSD and Benjamini-Hochberg corrections were used by one (3%) article each. One article did not specify any correction or adjustment for their post hoc analyses.

**Table 4.** Summary of the experimental designs across all articles and their corresponding analyses.

| Experimental Procedure | N | Analysis |
|---|---|---|
| Control/AR Design | 5 | *t*-test |
| | | One-way ANOVA |
| | | Wilcoxon Signed Rank Test |
| | | Pearson Correlation |
| | | Multiple Regression |
| x Age x distraction | 1 | Mixed ANOVA |
| x Reliability | 3 | *t*-test |
| | | Linear Mixed Effect Model |
| x Information Timing (during/app usage after) | 1 | Wilcoxon Signed Rank Test |
| | | Mann–Whitney Test |
| x Traffic density x Interaction complexity | 2 | Repeated Measures ANOVA |
| | | Cumulative Link Model |
| Control/2+ AR Designs | 6 | *t*-test |
| | | One-way ANOVA |
| | | Two-way ANOVA |
| | | Repeated Measures ANOVA |
| | | Cochran's Q |
| | | Friedman's Test |
| | | Kruskal–Wallis Test |
| | | Wilcoxon Signed Rank Test |
| | | NPAV |
| | | Pearson Correlation |
| x Congruent between maneuver and the situation | 1 | Mixed ANOVA |
| x Driving Scenario | 1 | Linear Mixed Effect Model |
| x Distance | 1 | Two-way Repeated Measures ANOVA |
| | | Durbin's Chi-square Test |
| AR Design/AR Design | 1 | Linear Mixed Effect Model |
| x Critical situation | 1 | *t*-test |
| x Visibility | 1 | Wilcoxon Signed Rank Test |
| | | Mann–Whitney Test |
| AR Design/2+ Designs | 2 | *t*-test |
| | | One-way ANOVA |
| | | Two-way Repeated Measures ANOVA |
| | | Pearson's Chi-square Test |
| x Distance | 1 | Two-way Repeated Measures ANOVA |
| x Urgency | 1 | Two-way Repeated Measures ANOVA |

**Table 4.** *Cont.*

| Experimental Procedure | N | Analysis |
|---|---|---|
| Display Modality Control/Tablet/AR x Visual Design | 2 | Repeated Measures ANOVA Friedman's Test NPAV |
| HDD/AR x Driving scenario | 1 | *t*-test Two-way Repeated Measures ANOVA |
| Baseline/HDD/AR | 1 | *t*-test Repeated Measures ANOVA |
| Alert modality (AR/Auditory) x Information type | 2 | Two-way ANOVA Friedman's Test Linear Mixed Effect Model Logistic Regression |

Note: N = total number of articles that included the experimental procedure design, total N includes two articles that conducted two studies. Bold = common experimental design, additional independent variables are noted by "x". AR = augmented reality. HDD = heads down display. ANOVA = Analysis of Variance. NPAV = Non-parametric Analysis of Variance. Baseline involves a minimal visual display; control is the absence of visual display.

### 3.1.8. Automation Level

Approximately over half of the articles (*n* = 18, 58%) involved vehicles equipped with some level of ADS features. Six (19%) utilized level 2 features; four (13%) utilized level 3 features; five (16%) utilized level 4 features, and three (10%) utilized level 5 features. Excluded from these counts are four articles: three did not explicitly state the level of automation features used or intended to use so were not included. One article stated that it was intended for vehicles equipped with level 4 and 5 features so was counted under level 4 ADS features. Another article conducted two experimental studies with the first study intended for vehicles equipped with levels 3–5 features so was counted under level 3 ADS features; the second study was intended for level 1 and 2 features so was counted under level 2 given the driving simulator engaged level 2 features.

As shown in Figure 3, research involving vehicles equipped with some form of automation features is increasing. This could be due to the technological improvements in driving simulators that allow some level of ADS features, or the increased use of Wizard-of-Oz methods. The general trend seems to place focus on operating with level 2 and level 4 automation capabilities. Given that level 2 (partial driving automation) driving automation system-equipped vehicles require drivers to monitor the environment, AR visualization research possibly supplements drivers' vigilance and environmental monitoring. In comparison, level 4 (high driving automation) ADS-equipped vehicles may represent AR visualization research for bringing the driver back in-the-loop and regain situation awareness quickly through the communication of relevant critical environment features or current ADS actions.

### 3.1.9. AR modality, Information Displayed, and Visualization

Overall, AR designs were presented using four different displays. Most prominently were AR windshield displays (*n* = 21; 68%) which presented information using the full windshield. Next, seven (23%) articles presented AR using optical see-through HUDs; an additional two (6%) utilized simulated optical displays. Finally, one article differentiated from displaying its information on a projected screen rather than on the windshield due to an artifact of the simulator set-up using a screen projector. A summary of the AR designs and modalities used for each study is shown in Table 1.

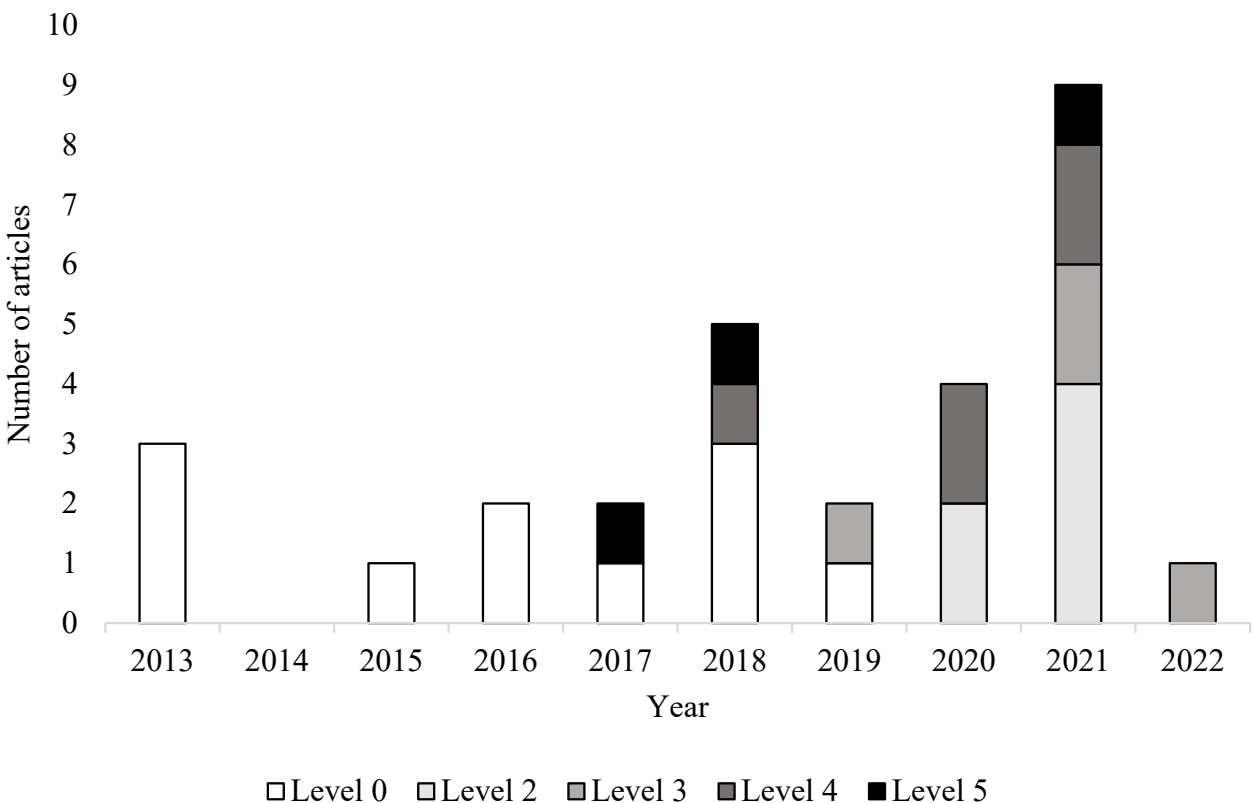

**Figure 3.** Automation level utilized in research articles by year.

Articles focused AR visualizations on seven different areas of communication. Twelve (39%) articles included pedestrian detection, with one further evaluating AR displays for pedestrian intention. Eleven (35%) evaluated vehicle detection displays. Most of these focused on vehicles in the general area while one specifically focused on merging vehicles and vehicles braking ahead. Ten (32%) evaluated hazard detection displays. Hazard detection includes construction sites, objects in the road, and other road actors such as cyclists. Articles that included the detection of vehicles and pedestrians as additional hazards, rather than the primary rationale of the display, were categorized under hazard detection. Ten (32%) evaluated AR displays that communicated the ADS's intended route or upcoming maneuvers. Five (16%) specifically focused on communicating nearby road signs. Two (6%) evaluated AR displays that communicated the ADS's reliability or confidence level of the current maneuver. Finally, only one (3%) article evaluated AR displays that communicated the vehicles' speed and upcoming speed change.

The most common AR visualization included bounding shapes around targets ($n$ = 11, 35%). Example visualizations are shown in Figure 4. Bounding shapes included boxes around targets ($n$ = 7), similar to Figure 4a, a dotted rhombus where the lines converge as the vehicle gets closer ($n$ = 2), and circles around the base of the target ($n$ = 2). These primarily concerned the detection of external objects such as pedestrians, vehicles, road signs, or other hazards. Ten (32%) articles utilized symbols and icons for object detection and pedestrian intention, similar to Figure 4b. Nine (29%) articles displayed arrows to primarily communicate the ADS's intended route. Seven (23%) articles used highlighting to focus attention for object detection. Highlighting was differentiated between solid object highlighting ($n$ = 5), similar to Figure 4c, and highlighting shadows under targets ($n$ = 2). Four (13%) articles utilized lane marking for intended route navigation, similar to Figure 4d, potential speed changes, and communicating the current level of automation reliability/confidence. Two (6%) articles presented lines in the environment to communicate vehicle and pedestrian detection as well as vehicles' merging direction. Only one (3%)

article presenting a lightband at the bottom of the windshield to highlight the direction of critical objects.

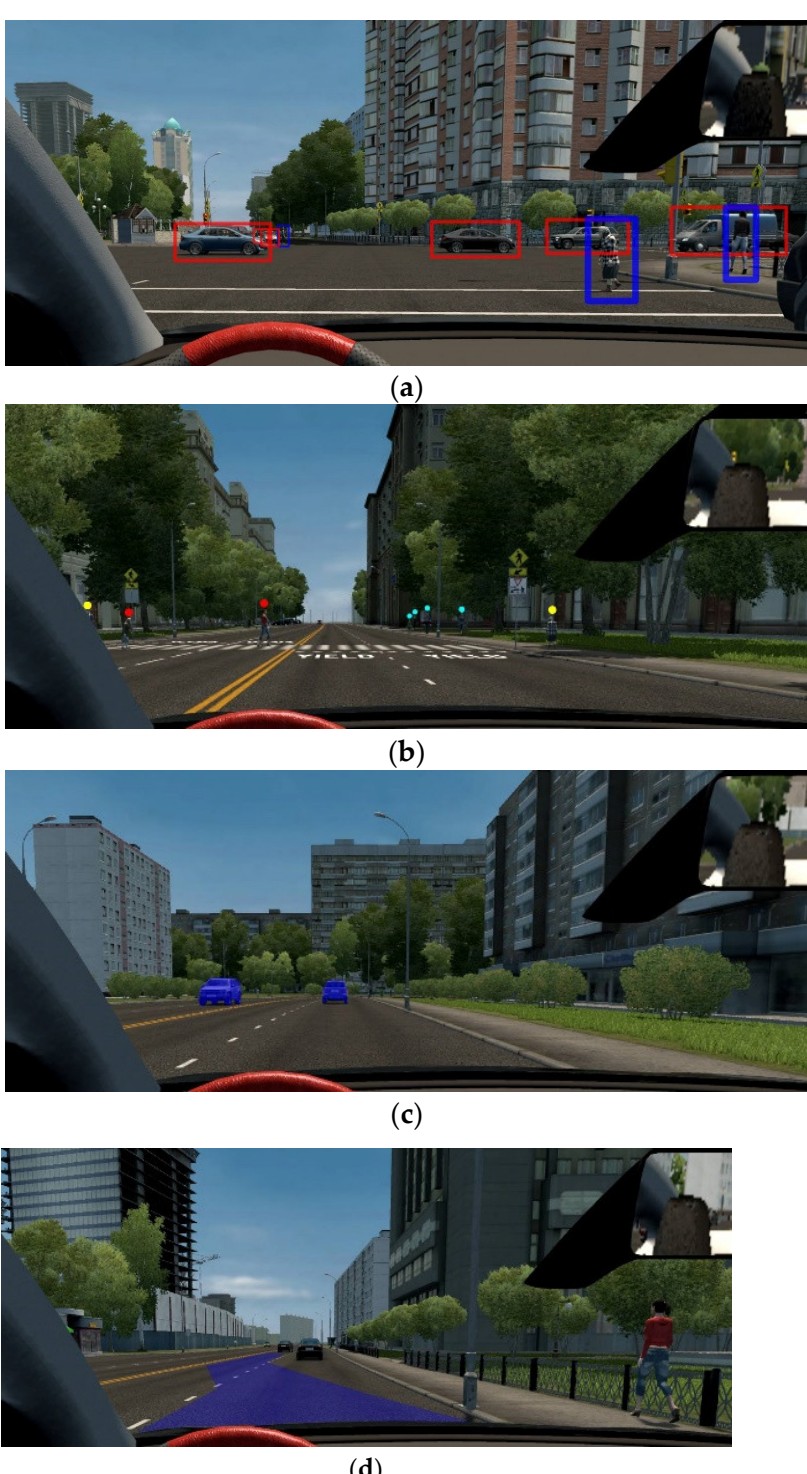

**Figure 4.** Example illustrations of four different windshield AR interface variations. (**a**) illustrates bounding boxes around pedestrians (blue) and vehicles (red). (**b**) illustrates color coded icons above pedestrians with three levels: pedestrian recognized (cyan), pedestrian intent to cross unclear (yellow), pedestrian crossing (red). (**c**) illustrates highlighting vehicles in blue. (**d**) illustrates a transparent blue lane trajectory indicating the vehicle's intended route due to a vehicle collision ahead.

*3.2. AR Visualizations–Key Results*

This section summarizes the key results of AR visualization interfaces across the different areas of communication. Although hazard, vehicle, and pedestrian detection were categorized as different areas, many of the articles evaluated displays that communicated in a combination of these areas and so are reviewed together according to AR visualization design. Therefore, this section proceeds as object detection (hazard, vehicle, and pedestrian detection differentiated by AR design), intended route, automation reliability, then speed.

3.2.1. Object Detection

Bounding Shapes

Early research found that a converging bounding rhombus design for object detection improved response times to warning signs, but not for vehicles with high visibility [68]. Using the same design, however, ref. [67] found no indication that this AR visual resulted in any additional benefit for drivers with lower attentional capacity compared to individuals with higher attentional capacity. Rather than a rhombus, ref. [66] found that presenting a converging bounding box helped participants perceive pedestrians faster and reduced the frequency of urgent braking.

More recent research evaluated bounding boxes with advantageous results. Compared to no AR cues, presenting bounding boxes (blue for pedestrians, red for vehicles) reduced the total number of AOIs scanned and number of glances before deciding to takeover manual control [58]. If participants decided to takeover, then takeover performance was much faster; if participants decided against resuming control, then the AR cues were related to longer glance duration and greater number of AOIs scanned. Interestingly, participants tended to glance more towards pedestrians than vehicles when AR cues were present, suggesting that AR cues potentially enhance situation awareness of pedestrians but not for vehicles. Ref. [74] evaluated similar AR visualizations and found that the benefit of presenting bounding boxes was mediated by different traffic conditions. That is, when there is greater pedestrian and vehicle density, more objects are highlighted in the visual field, subsequently, drivers' visual attention may shift away from the AR cues to try self-process the environment information.

Evaluating bounding boxes for broader hazards (e.g., pedestrians, vehicles, delivery persons, and construction sites) with color distinctions dependent on criticality, Ref. [70] found the presence of these AR cues led to significantly higher user experience scores, and greater understanding and ease-of-use of a level 5 ADS-equipped vehicle compared to the absence of the AR cues. Furthermore, participants rated AR visualizations consisting of bounding boxes and warning signs that communicated the ADS's intended action and its rationale as easy to understand [18]. However, participants preferred the AR visualization that also implemented verbal communication for better information-processing.

Finally, ref. [39] evaluated four AR visual designs and found that each design was better than no AR cueing for subjective measures of trust, acceptance, and perceived safety. The four designs including (a) a warning symbol, (b) a lightband at the base of the windshield that highlighted the direction of a hazard, (c) a bounding circle around a target, and (d) priority highlighting whereby individuals are highlighted by color-coded highlighting (red most critical, blue less critical, no visualization under a certain threshold). Across the four designs, the warning symbol and bounding circle were favored the most with higher usability scores compared to no AR cueing, and the lightband was least preferred with higher mental workload compared to no AR cueing and less comprehensible compared to the other visualizations.

Highlighting

In a pilot study, solidly highlighting both dynamic (vehicles and pedestrians) and static (road signs) on simulated footage was rated higher in subjective measures than highlighting dynamic objects alone, when displayed on a tablet, or without visual cues [53]. In their main evaluation, highlighting both dynamic and static objects on real-world footage led to

higher situation awareness without increasing cognitive load compared to no AR cueing. However, no differences were found for trust and usability assessments. When focusing on road signs, ref. [41] concluded that highlighting general road signs did not affect the detection of maneuver-related cues but did alter the allocation of visual attention while making decisions. When driver actions are required, they suggest highlighting relevant information rather than being overly general to optimize decision-making processes.

An AR display highlighting the predicted path of a pedestrian was better at pedestrian detection than without the AR; however, no difference was found between whether the visualization was fixed on the optical see-through HUD or in front of the pedestrian's location [38] and did not affect driver's allocation of visual attention [61]. However, ref. [61] found that visualizing pedestrians in bounding boxes attracted more visual attention to pedestrians but reduced attention to other environmental features such as unidentified vehicles, landmarks, or traffic signs. Additionally, although bounding boxes did not affect drivers' detection of environmental features such as road signs, the AR design did degrade drivers' ability for higher-level situation awareness to understand the meaning of the signs.

Compared to no AR cues, AR displays that either visualize an arrow atop of or shadow highlighting road actors significantly improved driving performance and effective eye gaze behavior [42]. That is, the presence of these AR visualizations effectively reduced the driver's visual distraction, improving the efficiency of obtaining visual information, and subsequently reduced response times to events. However, no significant differences were found between the two visualizations.

Highlighting hazards and anticipated vehicle maneuvers improves driving performance in scenarios that require an obvious steering reaction (e.g., ADS failure or upcoming construction site); however, more ambiguous situations lead to longer takeover reaction times (e.g., traffic rule ambiguity) [40]. Having a more complex AR display (involving highlighting pedestrians, traffic signals, and white navigation arrows using a simulated optical see-through HUD) did not improve driving performance in comparison to a display with a minimal design [55]. However, ref. [55] compressed road information into an isolated display area which may have increased visual clutter explaining their worsened SA findings compared to driving without AR visualizations. Furthermore, driving style led to different patterns of situation awareness across AR conditions. In contrast, presenting an aggregate of AR displays increased situation awareness [62] and resulted in higher trust than a heads-down display [64]. However, participants noted that continued presentation of information may become a negative aspect after familiarity.

Symbols and Icons

In an early study, ref. [59] utilized time-to-collision metrics using colored blocks at the base of leading vehicles which increase in number as time-to-collision decreases. This AR cue was effective in alerting drivers to merging traffic or when the lead vehicle was braking. Additionally, their results indicated that elderly drivers used the AR interface to maintain safer driving distances.

Ref. [43] utilized a comprehensive AR interface whereby symbol (triangle) orientation above vehicles identified either oncoming (upwards facing triangle) or preceding (downward facing triangle) direction with color coding to differentiate the ability to perform safe or unsafe actions via minimum-time-to-collision metrics. Results indicated that the presence of these AR cues led to increased trust and acceptance of the ADS-equipped vehicle as well as subjective comprehension of external vehicle actions especially in adverse weather conditions (i.e., dense fog). No differences in heartrate-variability were found between the presence of absence of AR cues [73].

Ref. [63] specifically evaluated four AR arrow visualizations via an optical see-through display for pedestrian detection in controlled, outdoor settings. The four displays included a screen-fixed static yellow pedestrian symbol indicating the direction and remaining distance to target; a conformal, world-fixed pedestrian crossing signpost located next to the pedestrian; an animated compass with a warning sign and an animated arrow

indicating pedestrian location; and an animated pedestrian icon with arrows indicating the predicted pedestrian path. Across the designs, no differences in visual attention or braking performance were found. However, the animated compass was less effective for appropriate vehicle stop gaps; consequent suggestions indicated that animations should be independent of the vehicle's motion if employing animated graphics.

Compared to specific auditory warnings, using specific warning symbols associated with specific hazards led to faster gaze responses [71], and improved driving performance such as reduced passed speed of hazards and number of collisions [72]. However, adding additional animation graphic effects did not improve driving performance. In the case of communicating an upcoming construction zone hazard, ref. [54] did not observe any clear trends between presenting more abstract (i.e., a warning symbol and/or the reasoning) or more specific (i.e., highlighting a traffic sign with bounding boxes) visualizations on cognitive load, acceptance, or question-response situation awareness. However, the situation awareness questions were unrelated to the hazard, for example, asking what the color of the nearby vehicle was or whether the participant passed a truck. Although those high in self-reported aberrant driving behaviors indicated higher mental workload and perceiving less information, these were only observed in a few conditions, but no overall pattern was observed.

Ref. [52] found that presenting warning symbols and icons via an AR windshield display was perceived more favorably than via a tablet device. Additionally, if communicating pedestrians' intent to cross, communicating three levels (intention to stay, intention to cross, intention unclear) led to reduce cognitive load and higher trust than only communicated whether intention is clear or unclear.

Lines

So far, practically all variations of AR visualizations have been shown to improve driving and eye-gaze behavior or enhance subjective ratings. However, ref. [60] found that placing a line under a target for detection did not significantly improve response time compared to the control group without AR cues.

3.2.2. Intended Route

Compared to no AR display, visualizing the ADS's intended route using blue transparent lane markings resulted in significantly greater understanding and ease-of-use of the ADS-equipped vehicle [70]. Additionally, lane markings were found to increase trust of the ADS-equipped vehicle compared to a heads-down display that presented similar visualizations [64].

Compared to a world-fixed conformal arrow that is displayed on the road ahead, a screen-fixed arrow for intended route navigation was associated with lower workload and higher usability [49]. Although there was no difference in objective driving measures, participants spent less visual attention toward the screen-fixed graphic suggesting better interpretability. Similarly, ref. [65] found that boomerang-shaped arrows were rated higher in clearness, interpretability, intuitiveness, than conformal solid arrows. However, tilting the arrow for added visual effect was detrimental to subjective ratings.

In contrast, ref. [56] found that, compared to screen-fixed arrow icons and a control condition (i.e., no visualization), an AR world-fixed arrow was associated with higher usability and greater trust of the vehicle system. Additionally, presentation of world-fixed arrows led to greater identification and appropriate participant reactions during system errors wherein the ADS failed to identify and obey a stop sign or intended to change lanes into an object. In this AR world-fixed arrow condition, participants gazed at the misperceived stop sign or object more often and reacted twice as fast as in the control condition, indicating that drivers may detect system failure earlier when clear ADS intentions are communicated.

Compared to a heads-down display, displaying a chain of arrows to indicate the ADS's intended route led to better steering reactions following TORs [40]. In the specific case

of a level 5 ADS-equipped vehicle, if passengers are facing the rear of the vehicle, then presenting an arrow of upcoming vehicle maneuvers on the rear windshield from the individual's perspective leads to better acceptance and trust rather than arrows presented from the vehicle's perspective [43]. Rear-facing perspective was better due to the possible mental effort required to rotate the visualization to fit the driving direction.

### 3.2.3. Automation Reliability

Two articles communicated ADS reliability to drivers. First, ref. [57] evaluated lane markings with varying colors distinguishing the reliability of the ADS against no AR visualizations. Results indicated that the presentation of AR lane markings led to faster takeover response times compared to no presentation. Furthermore, as the AR visualization communicated that the ADS reliability decreased, participants tended to increase glance duration on the road. Thus, suggesting that informing drivers of the systems' potential limitations induces monitoring of the road environment in case of possible TORs. Second, ref. [62] included an automation confidence bar in an aggregate of AR displays which overall increased SA, perceived safety, and trust; however, the inclusion of numerous other displays simultaneously did not indicate the impact of presenting ADS reliability.

### 3.2.4. Speed

Only one article evaluated AR interfaces for communicating upcoming speed changes. Ref. [69] compared lane marking with a moving horizontal bar to that of an arrow indicating increase or decreasing speed and found no clear advantage of one interface over the other. Of significance is that visualizing upcoming speed changes through the lane marking visual led to worsened situation awareness for the speed event.

## 4. Discussion

In this paper, we systematically identified 31 articles in the past decade that evaluated AR displays that communicated road elements or potential ADS actions to the driver. Throughout the review, we reported descriptives of high-level article information and aggregated AR visualizations across the articles. This section summarizes the main results and provides recommendations for the research and implementation of AR displays in ADS-equipped vehicles.

### 4.1. High-Level Descriptives

We found that more articles are being published within the past five years which coincides with the increased growth of technology within this area. Within these last five years, more conference articles were published which could be explained by the generally shorter article length and less time required for peer-review and revisions in comparison to journal articles. Articles originated mainly from Germany and the United States which is in line with these countries being two of the leading supporters of ADS-equipped vehicles [75].

Most of the research occurred in safe, controlled, laboratory settings using simulations of some kind. Although similar patterns of driving behavior are seen between driving simulators and naturalistic settings (e.g., [76,77]), ref. [53] did identify a different pattern of results when implementing the AR design in real-world footage as compared to simulated footage. However, differing patterns of results between the two settings was more identified when using optical see-through HUD rather than windshield displays. This distinction could be due to the optical display communicating all information in an isolated area, possibly increasing visual clutter as the road environment becomes more complex. However, more naturalistic, or at least controlled, outdoor research is required to evaluate the real benefits of AR communication as only three articles were conducted in more natural settings and eight simulator or online studies presented real-world footage.

Regarding participant information, most articles reported gender distribution and the mean age of participants. Approximately, half the articles reported the source of recruitment, yet only one article reported participants' ethnicity. Collectively, participants

tended to be young, healthy males which is not generalizable to the whole population. Only one article reported participants who did not self-identify as male or female. Two articles recruited individuals from a vulnerable population (i.e., elderly individuals) which resulted in different driving patterns to younger individuals when interacting with AR displays. Additionally, with AR visualizations using color coding schemes, no article mentioned accessibility issues to individuals with color blindness, though one article did specifically exclude any individual with self-reported color blindness. Therefore, greater transparency is recommended when reporting participant demographics but also the recruitment of diverse individuals such as individuals who identify as non-binary, neurodivergent individuals, or individuals from vulnerable populations. Greater transparency and diverse participant recruitment is required so that future designs are accessible across a more representative inclusive sample of the population.

### 4.2. AR Designs

Overall, there is a clear trend that communicating environmental elements and the ADS' actions is beneficial to drivers. Typically, the more favorable designs were those which presented clear, relevant information to the given context. In contrast, ambiguous or too much information led to worsened driving or situation awareness performance (see [55,60,69]). However, distinct design differences may play less strongly of a role as compared to the sole feature of presenting crucial information. Furthermore, the articles consistently found more favorable outcomes for AR displays than tablet or heads-down-displays. Research is still yet to compare optical see-through HUD displays and windshield displays. However, there is suggestion that optical HUD may have a threshold whereby too much visual clutter negates any decision-making or situation awareness benefits [55]. Across automation levels are apparent differences in why AR displays are needed. For instance, across all level features, presenting information may improve trust and acceptance of the ADS-equipped vehicle and dynamically calibrate appropriate expectations about the ADS's capabilities; however, for features operated at levels 2 and 3, there is an additional focus on enhancing drivers' situation awareness to improve takeover response times and safety concerns. At higher automation levels, situation awareness is less crucial due to the reduced need to resume manual control, or lack thereof, of the vehicle and can focus more on novel interactions and passenger experiences.

For object detection, bounding shapes and highlighting target cues tended to be more prominent across the research. Bounding shapes tend to be more limited compared to highlighting. For instance, visualizations bound pedestrians and vehicles, whereas, highlighting involved pedestrians and their predicted paths, vehicles, road signs, and the ADS's predicted path. However, ref. [39] found that participants preferred bounding visualizations rather than highlighting for object detection. Across the board, researchers found that displaying bounding shapes was better for communicating the ADS's detection of pedestrians than vehicles. Vehicles were considered highly salient in the environment, thus much easier to see regardless of the AR, but pedestrians and other targets (e.g., signs) were less salient which may be a better focus point in displays or even vehicles outside of the central point of road view.

Accordingly, AR displays should communicate relevant information rather than being overly general to improve driving behavior and crucial visual attention. Furthermore, some argued concerns that continuous presentation of information may become a negative aspect due to familiarity. Therefore, presenting only relevant information as they present into the drivers' visual field may mitigate these potential detrimental effects. One article did suggest an AR system that is capable of dynamically alerting drivers of road hazards only when the ADS detects that the driver is not already aware of them [58]. Alternatively, presenting information that requires a clear action by drivers such as intended ADS maneuvers resulting from an upcoming construction site or system failure.

The articles that evaluated multiple AR designs against a control group generally did not find significant improvements in visual attention or driving performance between the

AR groups. The lack of differences between AR design complexity indicates that more complex displays do not lead to more situation awareness; therefore, it is not necessary to pursue more eye-catching forms of AR displays. Rather, the advantages of AR communication could be due to simply presenting relevant road information which supplements drivers' decision-making or expectations of the ADS's capabilities.

AR cues can provide transparent communication regarding the reliability and confidence of the ADS, calibrating drivers' expectations and trust of the ADS' capabilities. Unfortunately, only two studies specifically included ADS reliability, though ref. [62] communicated reliability as part of an aggregated display. These displays utilized reliability as a percentage (i.e., 85% reliable). Ref. [57] visualized reliability through blue transparent lane markings and communicated the ADS's reliability for navigate upcoming maneuvers. Although not displaying ADS reliability, ref. [56] focused on participants' performance when presented with inappropriate ADS maneuvers due to system error (i.e., misperceiving stop signs or objects). Additionally, ref. [52] indirectly evaluated reliability through icons that communicated pedestrian intention. Reliability was indirectly presented through the "intention unclear" icon whereby the ADS could not confidently perceive the pedestrians' intention. Both lane marking and icons have initial support for communicating reliability for different actions with individuals identifying maneuverer errors quicker when presented with world-fixed arrows. Further research is required to garnish greater support.

### 4.3. Future Research

Throughout the systematic review, we found research gaps within this application area of AR in ADS-equipped vehicles. Identified are six key avenues suggested for future research including: user reporting and inclusive design; outdoor studies with AR; sub-optimal driving conditions; longitudinal impact; visual complexity, relevance, and clutter; and system reliability.

- User reporting and inclusive design. Reported within this review was the fact that the large majority of studies predominantly recruited young, healthy, white males. Recruiting largely this population group reduces generalizability to other populations and considerations of inclusive AR interface design. To ensure the advantages of transport mobility via ADS-equipped vehicles is received by all, research should evaluate interfaces recruiting individuals from vulnerable populations, those who are considered neurodiverse, and those who require greater visual accessibility requirements such as those who are colorblind.

- Outdoor studies with AR. Most of the research were conducted in safe, controlled laboratory settings. As AR technology in vehicles are advancing, there is strong motivation to progress research towards outdoor settings. This could be a progressive shift from laboratory studies to outdoor test tracks before on-road testing to understand how environmental and social factors influence the interaction between AR interfaces and drivers' behaviors and perceptions.

- Sub-optimal driving conditions. All but two articles evaluated AR displays during sunny, optimal conditions with clear visibility. The two articles that utilized AR interfaces during impaired visibility (i.e., foggy weather) driving situations found different effects than clear driving. As the technology for automated features advances and operation is less restricted to optimal conditions, there is a research avenue to understand how behaviors, attitudes, or reliance on the AR interface changes when road elements are lowest in visibility such as foggy or night-time driving.

- Longitudinal impact. The articles reviewed were one-off interactions or scenarios repeated within the same day. The reported results could thus be considered a possible artefact from novel interactions. As drivers will engage more and more with these emerging systems, further examination is required to understand how drivers' behaviors adapt over time as familiarity with the AR increases and expectations are dynamically calibrated.

- Visual complexity, relevance, and clutter. Many of the articles compared different visual AR complexity levels of the same design (e.g., highlighting dynamic or dynamic and static objects) or compared to a control group with no interface. As the visual complexity of an interface increases, so too does the risk of visual clutter. For example, an interface that highlights vehicles and pedestrians may risk visual occlusion and object sensitivity as the number of dynamic objects increases in the driving scene. Additionally, presenting too much information may direct drivers' attention away from the relevant, crucial information. Therefore, research should understand what information drivers find relevant in different contexts and the potential threshold of presenting too much information before the AR interfaces becomes detrimental.
- System reliability. The reviewed articles typically presented AR interfaces that had perfect reliability. Unfortunately, the current object detection techniques present in vehicles are not without error. Although a few articles examined communicating varying degrees of system reliability, more research is required to understand drivers' perceptions and behaviors during system failures where either the ADS fails to detect hazardous objects or communicates inappropriate maneuverers leading to detrimental outcomes.

## 5. Conclusions

In summary, the overall goal of this review was to provide a broad overview of user studies evaluating AR interfaces that communicated environmental elements or the ADS' actions and perceptions. Using a modified PRISMA method, we identified 31 relevant articles utilizing various levels of automated features, experimental design, methodologies, constructs, and AR interfaces. Many benefits of implementing AR interfaces in ADS-equipped vehicles were identified. In particular, AR displays generally improved driving performance through braking and takeover responses, improved allocation of visual attention towards the target without negatively impacting situation awareness of non-highlighted targets, and is positively perceived across trust, acceptance, and usability. This review serves to provide future researchers and practitioners of the current approaches used to evaluate AR interfaces and provide insight into the impacts of various interfaces on drivers' behaviors and perceptions. Although AR research is still in development, there are plenty of research avenues that require attention such as inclusive design, outdoor testing, longitudinal impacts, visual relevance and clutter, and system reliability. Regardless, it is strongly supported that integrated AR interfaces would lead to safer driving and higher trust and acceptance of ADS-equipped vehicles across all levels of automation features.

**Author Contributions:** Conceptualization, L.K. and Y.-C.L.; methodology, L.K.; formal analysis, L.K.; investigation, L.K.; data curation, L.K.; writing—original draft preparation, L.K.; writing—review and editing, L.K. and Y.-C.L.; visualization, L.K.; supervision, Y.-C.L. All authors have read and agreed to the published version of the manuscript.

**Funding:** This research received no external funding.

**Institutional Review Board Statement:** Not applicable.

**Informed Consent Statement:** Not applicable.

**Data Availability Statement:** Not applicable.

**Conflicts of Interest:** The authors declare no conflict of interest.

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
