# Peer review of "Augmented Reality for Vehicle-Driver Communication: A Systematic Review"

_safety, 2022_

Round 1

Reviewer 1 Report

The article describes current issues regarding the safe implementation of driving automation systems. The issues described in the article contain numerous references to the literature. The layout of the article has been well planned and guides the reader through the individual issues. Issues relevant to the proper use of augmented reality in vehicles have been described Augmented Reality for Vehicle-Driver Communication

The article collects information on the methods of using augmented reality described in the literature to inform vehicle drivers about road hazards and situation. The presented list included the number of articles on individual related issues, which can be treated as the degree of interest in selected ways of providing information to vehicle drivers.  

The article describes an interesting issue which is the use of AR methods in autonomous vehicles. These methods make it possible to provide information to the drivers of these vehicles in a way that minimizes their involvement and distraction. The article is a review article.  

The article uses and discusses in detail the method of systematic review of the literature on issues related to the broadly understood automation of transport. The article summarizes the knowledge and data collected so far and described in other articles in the field of transport automation. Information on separate articles, which provide information on specific topics, which include, but are not limited to: acceptance, user experience, situational awareness, and cognitive load have been collected and summarized in this article. Measures of these values are also presented.

  If the authors planned to use or indicate the selected AR method as the best in given conditions, it would be worth considering the use of the multi-criteria AHP selection method in the future.  

In my opinion, conclusions are consistent.  

the article requires re-verification in terms of references to literature sources. For example, in the first paragraph of the third chapter, an incorrect reference to a reference appears - "Error! Reference source not found". A similar situation occurs on page 13 (line 339).  

The title of Figure 3 should be corrected, as it does not correspond precisely to the content of the Figure.

Reviewer 2 Report

(1) The gaps among other relevant review articles are not clear, as indicated in section 1.1 by the authors, the current review focused on the specific designs of in-vehicle AR visualizations that communicate either road elements or the ADS’s actions and decision-making to the driver across levels of automated driving. However, it cannot be reflected in the title. Also, only 31 related papers were found in the last decade, which is few for a review article and suggests the target topic could be somehow narrow.

(2) Content organizations are not ideal for a reviewed article, e.g. no methodology summarizations were systematically reviewed, and no detailed objective and subjective evaluations and results were compared band analyzed. No specific experimental procedures were summarized and introduced.

(3) Section 4.3 lacks sufficient insights into the future trends regarding this topic, and the current wording was rough and general.

(4) No vivid illustrations were presented in this review article to help readers to understand this topic.

Reviewer 3 Report

The paper deals with systematic review of augmented reality fo vehicle-driver communication, which is very actual topic. Moreover, augmented reality in transportation is scientific field with a great potential, mostly in combination with other modern technologies (e.g. elements of autonomous mobility). The paper is systematic review, not a original scientific paper. I appreciate quite broad review with PRISMA application which can be very useful for the scientific papers in this scientific field. Very interesting is the Future Research (4.3), which can be more detailed. I consider this part as the key summarizing output (following the conducted review) for the further research.

Minor comment: I recommend to add some more detaiols to the part 4.3 and to the Conclusion. Two of the references are missing (Error - reference) – part 3. and 3.1.8.

Round 2

Reviewer 2 Report

Thank you for addressing my previous concerns, I think the manuscript is improved properly and will recommend it to be accepted.